# Numerical Investigation of the Hydrodynamic Behavior of Trash-Blocking Nets for Water Intake Engineering of Nuclear Power Plant

**Zhenqiang Jiang [1], Tongyan Wang [2], Bin Wang [1], Tiaojian Xu [2,*], Changlei Ma [3] and Kanmin Shen [1]**

[1] Key Laboratory of Far-Shore Wind Power Technology of Zhejiang Province, Hangzhou 311122, China; jiang_zq@hdec.com (Z.J.); binwangdut@outlook.com (B.W.); shenkanmin@163.com (K.S.)
[2] State Key Laboratory of Coastal and Offshore Engineering, Dalian University of Technology, Dalian 116024, China; wty6232@mail.dlut.edu.cn
[3] National Ocean Technology Center, Tianjin 300112, China; macl.norc@hotmail.com
* Correspondence: tjxu@dlut.edu.cn

**Abstract:** In order to ensure the safety of the cooling water source of coastal nuclear power plants (NPP), trash-blocking nets (TBNs) are usually installed at the entrance of the penstock to prevent marine sewage and organisms flowing into the front pool of the pump house of the nuclear power plant. The safety evaluation of these trash-blocking nets is of paramount importance for the stable operation of a nuclear power plant. However, there is no reliable analysis method for improving the design of trash-blocking nets and mooring systems. In this study, a numerical model of in-current trash-blocking nets based on the lumped mass method was developed to calculate the tension force on the trash-blocking nets and mooring system. A comparison with the experimental data indicates that the present numerical model is appropriate for calculating the in-current hydrodynamic loads on the trash-blocking nets. In addition, the effects of the width of trash-blocking nets, hanging ratio, water depth, and net solidity are discussed in detail, and the damage process of trash-blocking nets was also investigated. The results indicate that the maximum tension force on the trash-blocking net linearly increases with the increasing width of trash-blocking nets, and it is greatly decreased with the increase in the horizontal hanging ratio of trash-blocking nets. It can be increased by 200% when the net solidity is increased from 0.16 to 0.6. Two damage modes for mooring lines can be observed, which are determined by the strength of mooring lines.

**Keywords:** nuclear power plant; lumped mass method; flexible net structures; water intake; Morison equation; fluid–structure interaction

## 1. Introduction

Trash-blocking nets (TBNs), as shown in Figure 1, are structures located at the entrance of the open channel of coastal nuclear power plants (NPPs) to prevent the entry of marine debris which can damage hydraulic turbines. The damage of TBNs is not uncommon and can produce lots of problems related to the hydraulic turbine.

The report of "Intake Cooling Water Blockage" (WANO SOER 2007-2) published by the World Association of Nuclear Operators (WANO) analyzed 44 cold source blockage incidents in NPPs. Approximately 20% of the incidents directly affected the safety of water intake, and other incidents could cause a power reduction or even shutdown. In recent years, water intake blockage incidents at the CRUAS, Hongyan-he, Ningde, Ling'ao, and Qinshan nuclear power plants were also reported. With sudden outbreaks of jellyfish, seaweed, algae, and other marine organisms, the blockage of water intake engineering is becoming an important problem affecting the safety of nuclear power plants. Tang et al. [1] carried out optimization research on blockage prevention in water intake engineering and analyzed the hydraulic characteristics of shell debris in an open channel's water intake.

Wu et al. [2] analyzed the existing problems of a cold source seawater filtration system in a nuclear power plant, and established a sea biological interception filtration system. In general, the trash-blocking net system is a biological interception filtration system commonly used in the open channel of coastal NPP, which is beneficial for conducting the safety evaluation of trash-blocking nets to ensure the stable operation of the water intake engineering of the nuclear power plant.

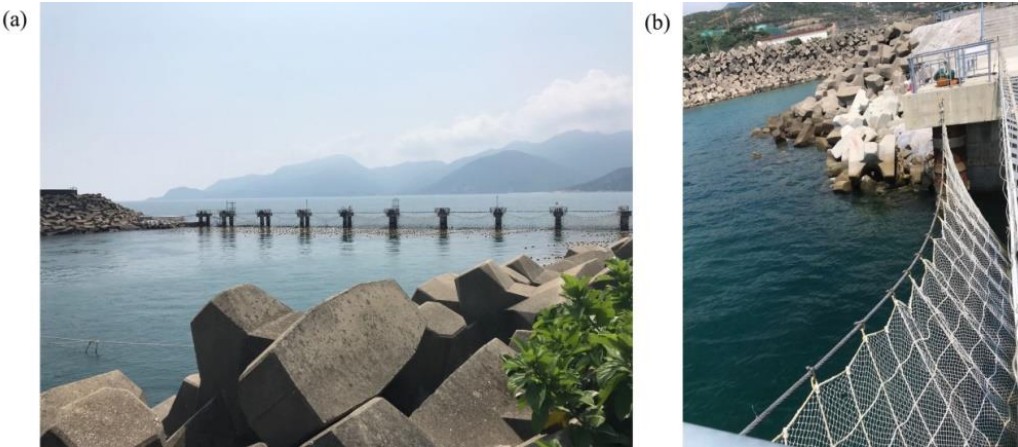

**Figure 1.** Trash-blocking nets attached at the pile foundations at the entrance of the open channel. (**a**) global view; (**b**) local view.

Flexible trash-blocking nets in coastal nuclear power plants will experience large deformation under the action of hydrodynamic loads, and thus, the analysis of the hydrodynamic response of trash-blocking nets in current is the basis of the design of trash-blocking nets. Many researchers have conducted lots of studies on the hydrodynamic response of flexible nets. Wan et al. [3] analyzed the deformation and tension distribution of a net set in a current based on the nonlinear finite element method. Tsukrov et al. [4] developed a numerical model to investigate the hydrodynamic response of net panels based on the consistent finite element method. Priour [5] proposed a finite element model for analyzing the flexible net structures, in which the triangular element was adopted to simulate the net. Takashi et al. [6,7] investigated the hydrodynamic loads and configuration of the bottom and drift gill net using the self-developed net-shape and loading analysis system. Lader et al. [8] developed a numerical model for 3D net structures in the combined waves and current, in which the super element method was adopted to simulate the nets. Balash et al. [9] measured the hydrodynamic loads on the plane net in current considering different mesh geometries, indicating that the drag coefficient of nets was a function of velocity, wave period, and the net's porosity. Kristiansen and Faltinsen [10] developed a screen type model for analyzing the hydrodynamic loads on nets, in which the net was divided into lots of net panels, or screens. Tang et al. [11] developed the fluid–net interaction model of a trawl system to analyze the hydrodynamic behavior of trawl, in which the k–$\omega$ shear stress turbulent model was used to simulate the flow field through the trawl, and the large deformation nonlinear structural model was applied to analyze the deformation of the trawl net. Cha et al. [12] derived the drag and lift coefficients of the chain-link woven copper alloys and knotless fabric nets, and visualized the flow through copper alloy net using particle image velocimetry (PIV). Zhan et al. [13] investigated the effects of the Reynolds number, net solidity, mesh pattern, and flow direction on the drag force on submerged nets. Gansel et al. [14] analyzed the flow regimes around nets with low and high values of solidity. In addition, Chan and Lee [15] analyzed the wave scattering due to a flexible net, in which a porous barrier was used to simulate the fish net. Kumar et al. [16] analytically investigated wave scattering by a flexible porous vertical membrane barrier in a two-layer fluid. Koley and Sahoo [17] investigated the interaction of obliquely incident waves with a vertical flexible permeable submerged membrane wave barrier. Patursson et al. [18,19] analyzed

the flow field around a net panel through computational fluid dynamics. Li et al. [20] developed a numerical model to analyze the shape and tension distribution of fishing nets in current based on the lumped mass method.

Large amounts of marine organisms can become attached to trash-blocking nets when these have been exposed in the open channels of coastal nuclear power plants for some time, and thus the solidity of a trash-blocking net and the drag force can be significantly increased due to the biofouling of the trash-blocking net. Several studies have focused on the drag coefficient of biofouled nets. Swift et al. [21] analyzed the drag coefficient of the clean and biofouled nets through the experimental test and the field measurements. The results indicated that the drag of the biofouled nets may be over three times that of clean nets. A drag coefficient of at least 0.6 (based on outline area) is recommended for the calculation of drag force on the biofouled nets. A considerable scatter for the drag force can be observed due to the different species present. Bi et al. [22] measured the drag force on the biofouled nets in currents through experimental tests, in which various levels of biofouling were considered. Cornejo et al. [23] analyzed the hydrodynamic effects of different levels of biofouling in fish cage aquaculture netting through numerical simulation, in which the clean and biofouled fish cage nettings were simulated using the porous media model.

The trash-blocking net is a biological interception filtration system commonly used in the open channels of coastal NPPs, and the calculation of the tension force on the trash-blocking net is an important basis of the design of trash-blocking nets. Although several numerical models have been proposed to simulate the flexible nets, previous studies have mainly focused on the hydrodynamic behavior of fish nets, especially for the fish cage system. Only very few studies have dealt with the hydrodynamic response of trash-blocking nets through experimental tests, and research on the tension force on the trash-blocking nets is greatly limited. Therefore, the numerical model of the trash-blocking net and mooring system in current is developed based on the lumped mass method for investigating the distribution of tension force on the trash-blocking net and mooring system. In addition, the influence of the width of the trash-blocking net, hanging ratio, water depth, and the level of biofouling on the tension force on the trash-blocking net and mooring system is discussed in detail. The damage mechanism of the trash-blocking net and mooring system is also analyzed, which is important for the optimal design of the trash-blocking net and mooring system.

This study is organized as follows: the numerical model of the trash-blocking net and mooring system is introduced in Section 2; the experimental validation of the nets in current is conducted in Section 3; after that, the distribution of the tension force on the trash-blocking net and mooring system in current is investigated in Section 4; finally, some conclusions are given in Section 5.

## 2. Numerical Methods

Trash-blocking nets have a flexible porous structure which can experience large deformation under the action of hydrodynamic loads. As a permeable structure, the trash-blocking net has relatively small influence on the flow field compared to other porous coastal structures, such as rock-mound breakwater. In addition, the diameter of the net twine is very small, and thus it is reasonable to calculate the hydrodynamic load on the net using the Morison equation. Therefore, the lumped mass method was adopted to simulate the hydrodynamic response of the trash-blocking net and mooring system in this study.

### 2.1. Structural Model of Trash-Blocking Net

The trash-blocking net is composed of a series of cylinders. In our study, the lumped mass method was applied to simulate the trash-blocking net, in which the trash-blocking net was simplified as a series of mass points connected by massless springs, as shown in Figure 2.

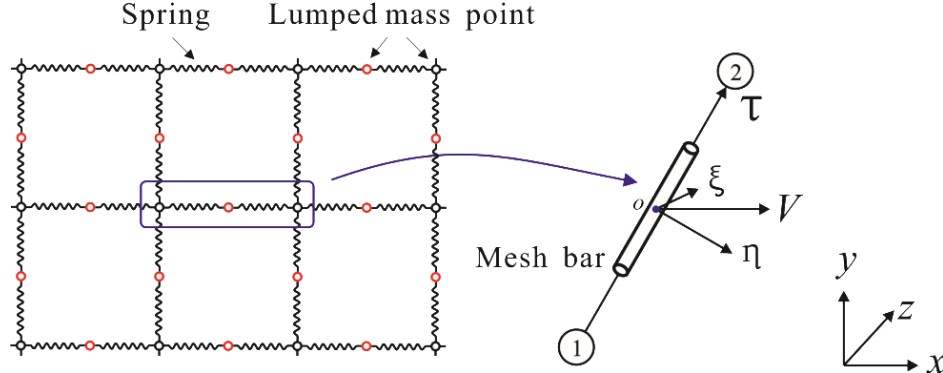

**Figure 2.** Schematic diagram of the trash-blocking net model.

According to Newton's second law, the motion equations for each node of the trash-blocking net can be obtained by

$$m_i a_i = \sum_{j=1}^{count} (T_j + W_j + B_j + F_{Dj} + F_{Ij})$$  (1)

where subscript $i$ denotes the node's sequential number; subscript $j$ denotes the associated neighboring element; count denotes the total number of neighboring elements to node $i$; $T$ is the tension force on net twine; $W$ is the gravity force; $B$ is the buoyancy force; $F_D$ and $F_I$ are the drag and inertia forces, respectively. Gui et al. [24] indicates that the inertial force $F_I$ on the fish net is rather small compared to other external forces, and thus it is omitted here.

In order to calculate the hydrodynamic force on the net, a local coordinate system $O\text{-}\tau\eta\xi$ is defined with an origin at the center of the mesh bar, and the Morison equation is adopted to calculate the hydrodynamic force on the net.

$$F_{D\tau} = \frac{1}{2}\rho C_{D\tau} D l \left| \vec{V}_\tau - \dot{\vec{R}}_\tau \right| (\vec{V}_\tau - \dot{\vec{R}}_\tau)$$  (2)

where $C_{D\tau}$ is the drag coefficient; $D$ is the diameter of net mesh bar; $l$ is the mesh size; $\rho$ is the density of water; $\dot{\vec{R}}$ and $\vec{V}$ are the velocity of the lumped-mass point and water particle, respectively. A similar expression can also be used to calculate the hydrodynamic force along the $\eta$ and $\xi$ axis directions.

Referring to the studies on the hydrodynamic resistance of towed cables in Choo and Casarella [25] and the hydrodynamics of a submersible net cage in Xu et al. [26], the hydrodynamic coefficient of the net mesh bar is related to the Reynolds number, which can be calculated as follows

$$C_n = \begin{cases} 8\pi\left(1 - 0.87s^{-2}\right)/(\text{Re}_n s) & (0 < \text{Re}_n \leq 1) \\ 1.45 + 8.55\text{Re}_n^{-0.90} & (1 < \text{Re}_n \leq 30) \\ 1.1 + 4\text{Re}_n^{-0.50} & (30 < \text{Re}_n \leq 10^5) \end{cases}$$  (3)

$$C_\tau = \pi\mu(0.55\text{Re}_n^{1/2} + 0.084\text{Re}_n^{2/3})$$  (4)

where $\text{Re}_n = \rho V_{Rn} D/\mu$, $s = -0.077215665 + \ln(8/\text{Re}_n)$; $\mu$ is the viscosity coefficient of water; $C_n$ and $C_\tau$ are the normal and tangential drag coefficients for the net mesh bar, respectively; and $V_{Rn}$ is the normal component of the fluid velocity relative to the mesh bar. According to Fredheim and Faltinsen [27], the knot part can be simulated as a sphere, and the drag coefficient for the knot part was set as 1.0 in this study.

The tension force on the net twine can be calculated as a function of the elastic elongation of the net twine based on Wilson [28] as follows

$$T = d^2 C_1 \varepsilon^{C_2}, \; \varepsilon = \frac{l - l_0}{l_0} \tag{5}$$

where $T$ is the tension force; $l_0$ is the initial length of net twine; $l$ is the deformed length; and $d$ is the diameter of net twine. Referring to Gerhard Klust [29], $C_1$ and $C_2$ are set as $345.37 \times 10^6$ and $1.0121$, respectively. More details on the numerical model of the flexible net in current can be found in Xu et al. [30].

### 2.2. Numerical Setup

The trash-blocking net is connected to the pile foundation by a series of mooring lines, as shown in Figure 3. The tension force on the trash-blocking net and mooring system in current is analyzed. The sequence number of mooring lines is given for the sake of clarity. The height and width of the trash-blocking net are 11.3 m and 15.0 m, respectively. The length of the net mesh bar is 50.0 mm whilst the diameter of the net twine is 3.5 mm. As a permeable structure, the influence of nets on the flow field was neglected in this study, the current velocity is equal to 0.2 m/s and the water depth is 7.0 m.

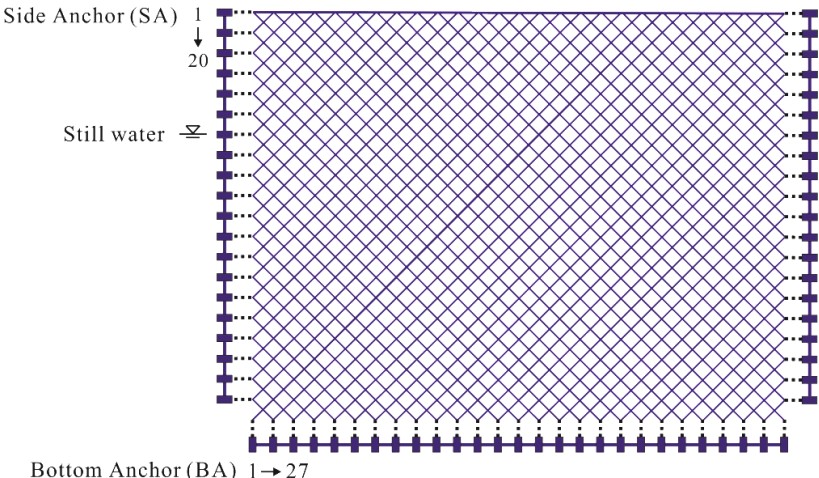

**Figure 3.** The trash-blocking net and mooring system for the water intake in the NPP.

### 3. Experimental Validation

To validate the numerical model of the trash-blocking net and mooring system in current, a series of experimental tests were conducted in the water tank (69 m long, 2 m wide, and 1.8 m deep) of the State Key Laboratory of Coastal and Offshore Engineering in Dalian University of Technology.

### 3.1. Plane Net Panel with Four-Point Mooring System

The trash-blocking net is connected to the pile foundation by four mooring lines, as shown in Figure 4. The tension forces on the mooring lines are recorded by the load cells attached to the mooring lines. The trash-blocking net is 0.73 m high and 1 m wide, and the length and diameter of the net mesh bar are 1.5 cm and 1.5 mm, respectively. Two types of water depths ($D = 0.6$ m for the high water level and $D = 0.47$ m for the low water level) are considered here. The high water level velocity and low water level velocity are 4.9 cm/s and 6.5 cm/s, respectively.

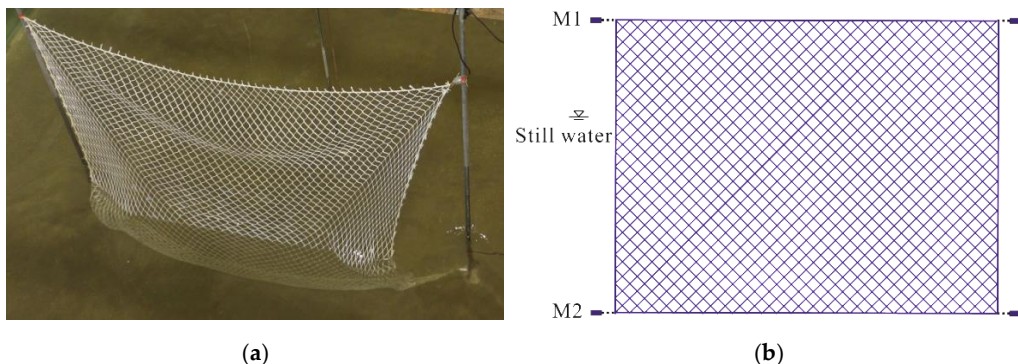

(**a**)  (**b**)

**Figure 4.** Setup of a net panel and four-point mooring system in current (**a**) experimental test; and (**b**) numerical simulation.

Figure 5 shows the in-current deformation and tension distribution of trash-blocking nets. The results indicate that the top part of a trash-blocking net experiences the largest tension force in this situation, a large deformation can be observed, and the shape of the trash-blocking net from the numerical simulation is similar to the experimental data (refer to Figure 4).

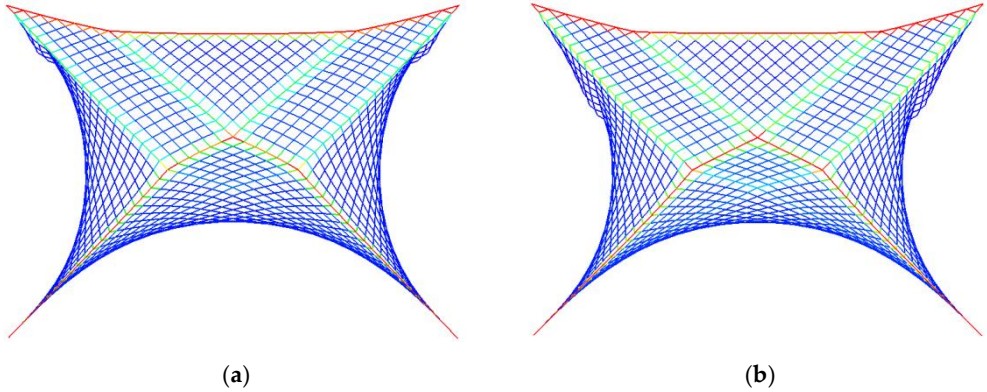

(**a**)  (**b**)

**Figure 5.** Deformation and tension distribution of trash-blocking net in current: (**a**) high water level; and (**b**) low water level.

In addition, the maximum tension forces on the mooring lines (M1 and M2 in Figure 4) are calculated and compared with the experimental data, as shown in Table 1. The results indicate that the relative error of the maximum tension force on the mooring lines between numerical results and experimental data is smaller than 7.0%, and the present numerical model is appropriate for calculating the in-current tension force on mooring line of the trash-blocking net. In general, the hydrodynamic load on the nets is related to the area of the net under the water surface and the square of velocity, and the water intake for the high water level and low water level is the same, and thus; it is therefore reasonable that the maximum tension force on mooring lines for the low water level is larger than that for the high water level.

**Table 1.** Maximum tension force on mooring lines for the trash-blocking net in current.

| No. | High Water Level | | | Low Water Level | | |
|---|---|---|---|---|---|---|
| | Exp. (N) | Num. (N) | Err. | Exp. (N) | Num. (N) | Err. |
| M1 | 0.221 | 0.234 | 5.9% | 0.261 | 0.270 | 3.4% |
| M2 | 0.121 | 0.129 | 6.6% | 0.162 | 0.166 | 2.5% |

*3.2. Plane Net Panel with Eight-Point Mooring System*

Because it is difficult to accurately measure the tension force on the nets, the tension force on the mooring lines is determined by the distribution of tension force on the nets. To further validate our numerical model of the net panel in current, the tension force on the mooring lines for the net panel with the eight-point mooring system is measured in this part. As shown in Figure 6, six points on the top of the net were fixed on a steel rod (1.5 m long) on the water surface and the two bottom corners were fixed to two sinkers. The width and height of the net panel are 1.5 m and 1.8 m, respectively. The mesh size is 40 mm, and the diameter of net twine is 3.5 mm. The mesh numbers along the width and height directions are 60 and 60, respectively. The horizontal hanging ratio is 0.64. The water depth is set to 1.5 m, and the flow velocity is set to 5 cm/s, 10 cm/s, 15 cm/s, and 20 cm/s. Three strain gauges were adopted to record the tension force on the mooring lines.

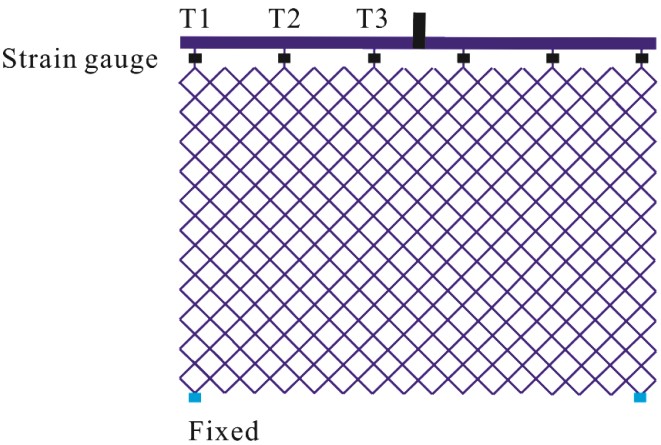

**Figure 6.** Setup of net panel and eight-point mooring system.

Figure 7 shows the in current tension force on the mooring lines for the net panel from numerical simulation and physical model test. The average relative error between the numerical result and experimental data is 6.8%. This indicates that the numerical model is appropriate for calculating the tension force on the net panel and mooring lines.

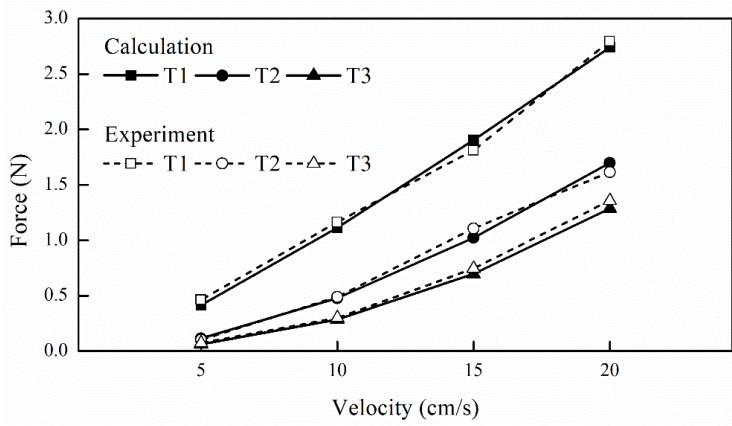

**Figure 7.** Tension force on the mooring lines from the numerical simulation and physical model test.

The ratios of tension force at each mooring line to the sum of tensions on the rod are described as $R_1$, $R_2$, and $R_3$, such as: $R_i = T_i(T_1 + T_2 + T_3)$, $i = 1, 2, 3$. Figure 8 shows the ratios of tension force for $T_1$, $T_2$, and $T_3$. $R_1$ is larger than the other two values in all cases, and this indicates that the load is concentrated on the outside top of the net. Figure 9 shows that the distribution of tension force on the net panel with an eight-point mooring system, and the results indicate that the tension force on the net is concentrated at the diagonals of

the net, especially for the small velocity case. The distribution of the tension force on the net panel with an eight-point mooring system is similar to the result in Suzuki et al. [31]. Therefore, it is reasonable that the tension force on the strain gauge $T_1$ is significantly larger than that on $T_2$ and $T_3$.

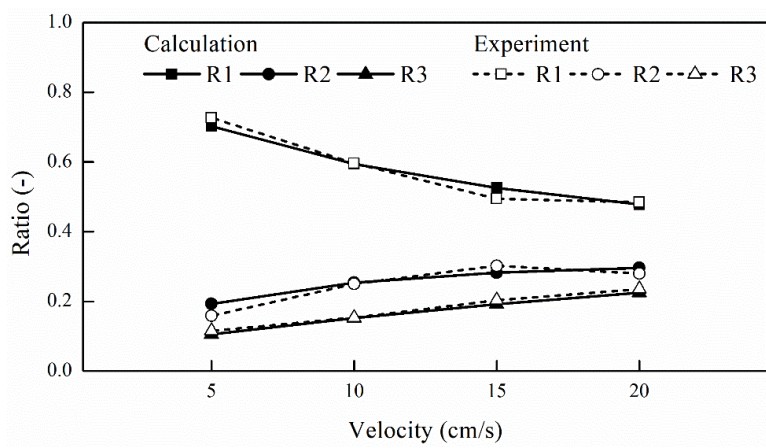

**Figure 8.** Ratios of tension force on the mooring lines from numerical simulation and physical model test.

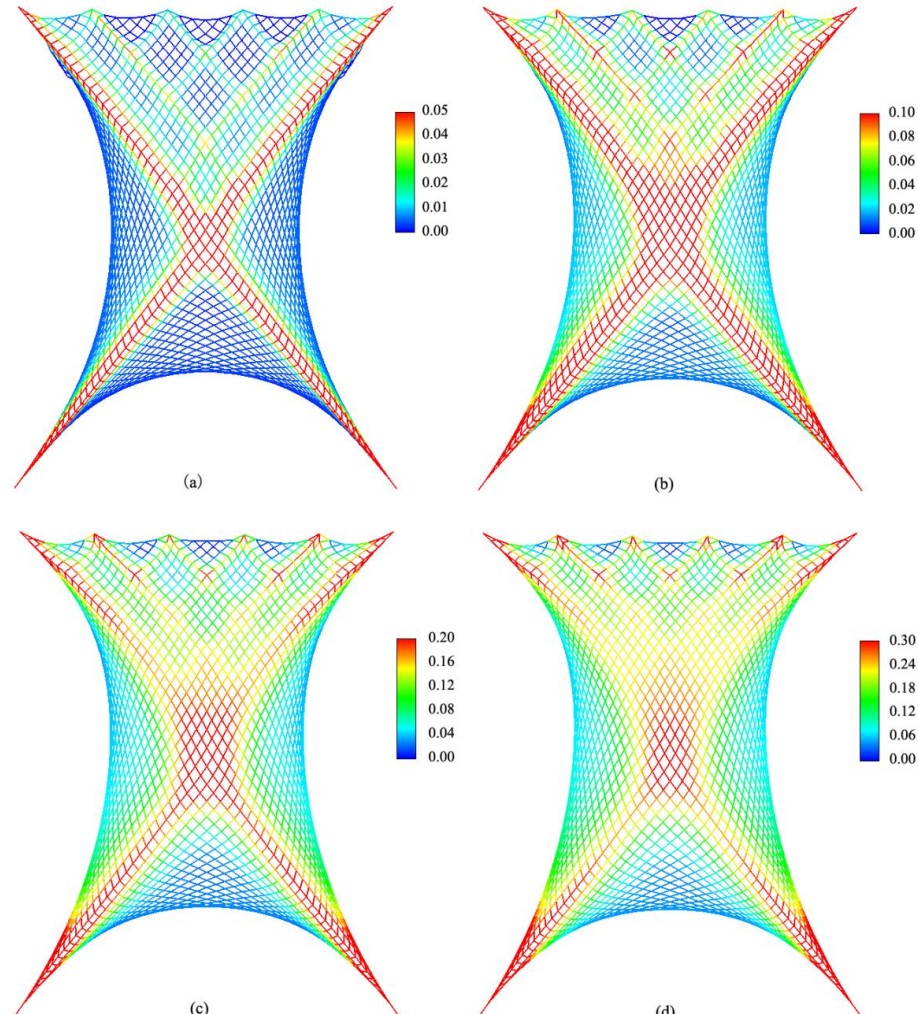

**Figure 9.** Three-dimensional graphic of the distribution of tension force on nets in current (unit: N): (**a**) U = 0.05 m/s; (**b**) U = 0.10 m/s; (**c**) U = 0.15 m/s; and (**d**) U = 0.20 m/s.

## 4. Results and Discussion

The in-current tension force on the trash-blocking net and mooring system is analyzed. The effect of the width of the trash-blocking net, hanging ratio, water depth, and level of biofouling is discussed in detail, and the damage process of the trash-blocking net and mooring system in the storm was also investigated.

### 4.1. Effect of the Width of Trash-Blocking Nets

The trash-blocking net is attached to the pile foundation, and the increasing width of the trash-blocking net can reduce the number of pile foundations in the open channel of nuclear power plant. However, the increasing width of the trash-blocking net will affect the tension force on the trash-blocking net and mooring system. In this part, the influence of the width of the trash-blocking net on the tension force on the trash-blocking net and mooring system is investigated.

Three kinds of widths of trash-blocking nets ($W$ = 15 m, 18 m, and 24 m) are considered, and the horizontal hanging ratio of the net is set to 0.724. Figure 10 shows the distribution of the tension force on the trash-blocking net. This indicates that the top part of the trash-blocking net is the most loaded. The maximum tension force on the trash-blocking nets for $W$ = 15 m, 18 m, and 24 m is 5.2 N, 8.5 N, and 15.6 N, respectively. The maximum tension force on the trash-blocking nets linearly increases with the increasing width of net.

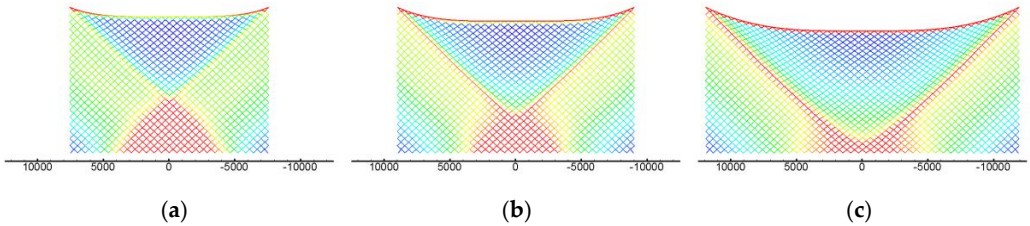

**Figure 10.** Distribution of tension force on the trash-blocking nets for different net widths: (**a**) $W$ = 15 m; (**b**) $W$ = 18 m; and (**c**) $W$ = 24 m.

The tension force on mooring lines is important for the design of trash-blocking nets and mooring systems, and thus, the tension force on the side moorings and bottom moorings is investigated herein. Figure 11 shows the maximum tension force on the side moorings for $W$ = 15 m, 18 m, and 24 m. This indicates that the No. 1 side mooring is most loaded, and the maximum tension force on Nos. 2–20 side moorings is significantly smaller than that of the No. 1 side mooring. The maximum tension force on the side moorings for $W$ = 15 m, 18 m, and 24 m is 81.6 N, 133.9 N, and 246.9 N, respectively; and it is linearly increases with the increase in the width of the trash-blocking net. Figure 12 shows the maximum tension forces on the bottom moorings for $W$ = 15 m, 18 m, and 24 m. This indicates that the maximum tension force on the bottom moorings gradually increases from the side part to the middle part. The maximum tension force on the bottom moorings for $W$ = 15 m, 18 m, and 24 m is 34.1 N, 38.2 N, and 43.0 N, respectively. This linearly increases with the increase in width of trash-blocking nets.

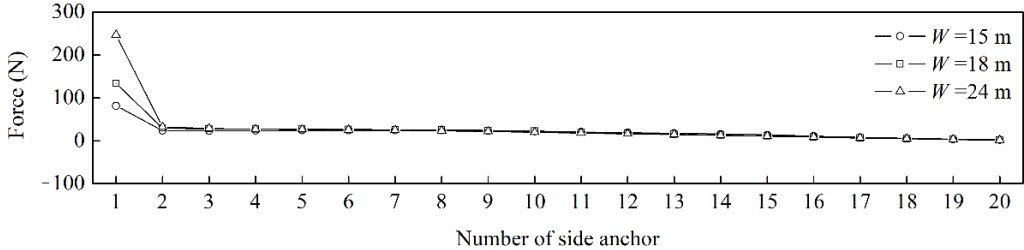

**Figure 11.** Maximum tension force on the side moorings for different trash-blocking net widths.

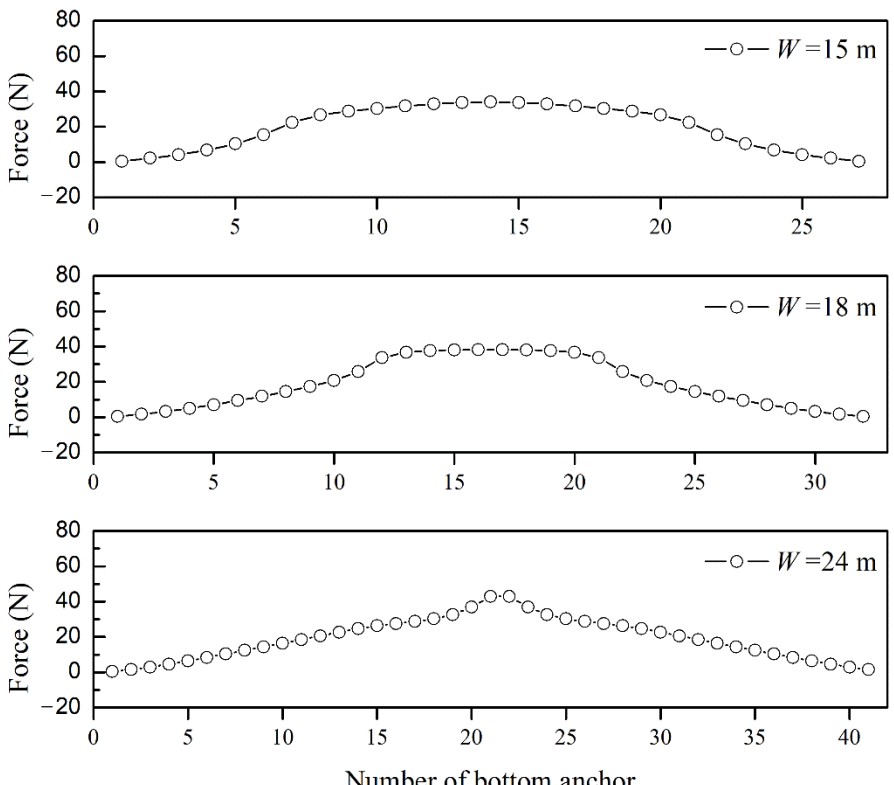

**Figure 12.** Maximum tension force on the bottom moorings for different trash-blocking net widths.

The total forces on the side moorings and the bottom moorings are important for the design of pile foundations. The total force on the side moorings for $W$ = 15 m, 18 m, and 24 m is 412.1 N, 471.2 N, and 571.8 N, respectively. The total force on the bottom moorings for $W$ = 15 m, 18 m, and 24 m is 526.8 N, 603.2 N, and 780.1 N, respectively. This is linearly increased with the increasing width of trash-blocking nets.

### 4.2. Effect of the Hanging Ratio of Trash-Blocking Nets

The tension force on the trash-blocking net and mooring system is also affected by the hanging ratio of the nets, and thus, six types of hanging ratios are considered here, as shown in Table 2. As shown in Figure 13, the horizontal ($R_h$) and vertical ($R_v$) hanging ratios of the net panel are equal to 0.5 $l_h/a$ and 0.5 $l_v/a$, respectively. In addition, the comparison of the tension force between the diamond net and the square net is also conducted.

**Table 2.** Tension force on the trash-blocking nets for different hanging ratios.

| No. | Mesh Type | $R_h$ | $R_v$ | $F_N$ | $F_{SM}$ | $F_{TSM}$ | $F_{BM}$ | $F_{TBM}$ |
|-----|-----------|-------|-------|-------|----------|-----------|----------|-----------|
| 1 | Diamond | 0.3535 | 0.9354 | 20.0 | 310.5 | 469.7 | 23.4 | 669.5 |
| 2 | Diamond | 0.4730 | 0.8810 | 14.6 | 227.2 | 418.5 | 34.2 | 608.2 |
| 3 | Diamond | 0.7070 | 0.7070 | 5.5 | 86.0 | 419.8 | 34.9 | 544.5 |
| 4 | Diamond | 0.8826 | 0.4701 | 2.6 | 28.8 | 708.6 | 20.4 | 385.4 |
| 5 | Diamond | 0.9191 | 0.3940 | 2.6 | 17.8 | 888.2 | 17.7 | 302.9 |
| 6 | Square | - | - | 2.3 | 37.2 | 594.4 | 3.4 | 91.2 |

$R_h$ is the horizontal hanging ratio; $R_v$ is the vertical hanging ratio.

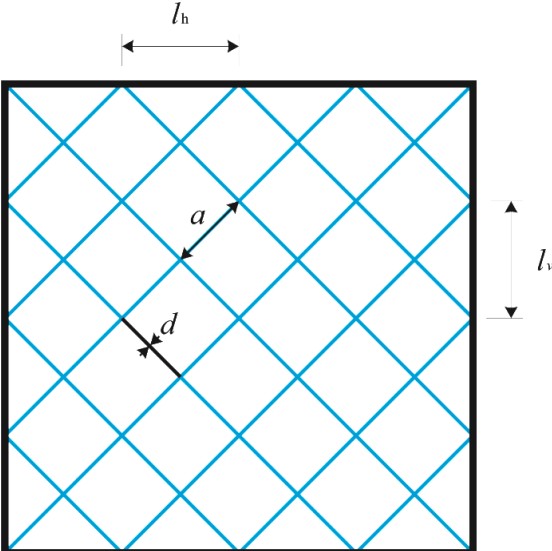

**Figure 13.** Description of solidity and hanging ratio of net panel.

Figure 14 shows the tension force on the trash-blocking nets ($F_N$) for different hanging ratios. The results indicate that the maximum tension force on the trash-blocking net is significantly affected by the load transfer path for different hanging ratios. Table 2 presents the maximum tension force on the trash-blocking net and mooring system for different hanging ratios. The results indicate that the maximum tension force on the trash-blocking nets is significantly decreased with the increase in the horizontal hanging ratio, and it is decreased by 87% when the horizontal hanging ratio is increased from 0.3535 to 0.9191.

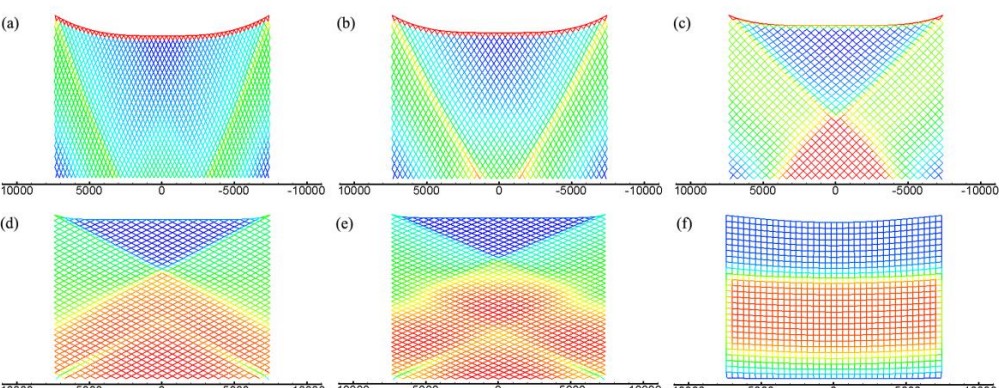

**Figure 14.** Distribution of the tension force on the trash-blocking nets for different hanging ratios: (**a**) $R_h = 0.3535$; (**b**) $R_h = 0.4730$; (**c**) $R_h = 0.7070$; (**d**) $R_h = 0.8826$; (**e**) $R_h = 0.9191$; and (**f**) Square mesh.

In addition, the maximum tension force on the side moorings ($F_{SM}$) and the bottom moorings ($F_{BM}$) is also significantly decreased with the increasing horizontal hanging ratio. With regard to the total force on the bottom moorings and the side moorings, the total force on the bottom moorings ($F_{TBM}$) is greatly decreased with the increasing horizontal hanging ratio, and the total force on the side moorings ($F_{TSM}$) is smallest for the medium horizontal hanging ratio ($R_h = 0.707$). Therefore, it is beneficial to select a medium horizontal hanging ratio when the trash-blocking net is installed in the pile foundation. The results also indicate that the tension force on the square net is greatly smaller than that of the diamond net.

### 4.3. Effect of Water Depth

The water intake from the open channel is of great importance for the safety of the coastal nuclear power plant, and it should be constant to ensure the stable operation of

nuclear power plant; however, the water level in the open channel can be varied. Therefore, the influence of the water level on the hydrodynamic response of the trash-blocking net and mooring system is analyzed herein. Seven types of water depth (*D*) are considered in this part, as shown in Table 3. In this part, the quantity of water flowing through the trash-blocking net is set to be same for different water depths, and can be calculated as $Q = W \cdot D \cdot U = 21 \text{ m}^3/\text{s}$. The horizontal hanging ratio of trash-blocking nets ($S_h$) is 0.707.

**Table 3.** Maximum tension on trash-blocking net and mooring system for different water depths.

| No. | *D* (m) | *W* (m) | *U* (cm/s) | $F_N$ | $F_{SM}$ | $F_{TSM}$ | $F_{BM}$ | $F_{TBM}$ |
|-----|---------|---------|------------|-------|----------|-----------|----------|-----------|
| 1 | 4.0 | 15.0 | 35.0 | 5.0 | 40.7 | 593.4 | 57.6 | 883.8 |
| 2 | 5.0 | 15.0 | 28.0 | 4.3 | 44.8 | 507.7 | 49.6 | 723.8 |
| 3 | 6.0 | 15.0 | 23.3 | 4.2 | 65.2 | 462.2 | 43.0 | 628.8 |
| 4 | 7.0 | 15.0 | 20.0 | 5.5 | 86.0 | 419.8 | 34.9 | 544.5 |
| 5 | 8.0 | 15.0 | 17.5 | 6.3 | 98.7 | 394.6 | 29.5 | 494.1 |
| 6 | 9.0 | 15.0 | 15.5 | 7.1 | 110.1 | 366.7 | 24.5 | 438.8 |
| 7 | 10.0 | 15.0 | 14.0 | 7.6 | 117.9 | 341.5 | 21.4 | 389.0 |

Table 3 presents the maximum tension force on the trash-blocking net and mooring system for different water depths. The results indicate that when the water depth is 6.0 m, the maximum tension force on the trash-blocking nets is smallest. With the increase in water depth, the maximum tension force on the side mooring ($F_{SM}$) is increased, and the maximum tension force on the bottom mooring ($F_{BM}$) is decreased. When the water depth is increased from 4.0 m to 10.0 m, the total forces on the side moorings ($F_{TSM}$) and the bottom moorings ($F_{TBM}$) are decreased by 42.5% and 55.9%, respectively. In general, a medium water depth is beneficial for the safety design of trash-blocking nets and mooring system.

### 4.4. Effect of Level of Biofouling

For the trash-blocking nets, the most serious loading comes from the trashes falling at the net, and thus, the hydrodynamic response of biofouled trash-blocking nets is analyzed here. In this part, five types of net solidities are considered to represent the effect of biofouling at trash-blocking nets. According to Swift et al. [13], a drag coefficient of 0.6 is adopted to calculate the drag force on the trash-blocking net based on the outlined area of trash-blocking net. The water depth and velocity are set to 7.0 m and 0.2 m/s, respectively. The horizontal hanging ratio of trash-blocking nets ($S_h$) is 0.707.

The maximum tensions on trash-blocking nets and mooring systems for different solidities are shown in Table 4. The results indicate that the maximum tension force on the trash-blocking net is increased by 200% when the net solidity is increased from 0.14 to 0.6, and the maximum tension force on the side moorings and the bottom moorings is increased by 202% and 220%, respectively. In general, it is important to clean the trashes falling on nets for reducing the maximum tension on the trash-blocking net and mooring system.

**Table 4.** Maximum tension on trash-blocking net and mooring system for different net solidities.

| No. | *d* | *Solidity* | $F_N$ | $F_{SM}$ | $F_{TSM}$ | $F_{BM}$ | $F_{TBM}$ |
|-----|------|------------|-------|----------|-----------|----------|-----------|
| 1 | 5.0 | 0.2 | 10.4 | 162.7 | 815.5 | 68.1 | 1085.7 |
| 2 | 7.5 | 0.3 | 13.0 | 204.6 | 1034.5 | 86.6 | 1388.2 |
| 3 | 10.0 | 0.4 | 14.9 | 233.6 | 1189.0 | 99.7 | 1606.1 |
| 4 | 12.5 | 0.5 | 16.0 | 251.9 | 1287.5 | 108.0 | 1749.8 |
| 5 | 15.0 | 0.6 | 16.5 | 260.1 | 1334.1 | 111.9 | 1823.8 |

### 4.5. Damage Process of Trash-Blocking Nets

The damage of the trash-blocking net and mooring system may occur when it suffers from the typhoon storm. In this part, the current velocity is set to 1.0 m/s, and the water depth is 7.0 m. The horizontal hanging ratio of net is 0.707, and the net solidity is 0.6. Figures 15 and 16 show the damage process of the mooring system of the trash-blocking net in the typhoon storm. Two types of damage modes of the mooring system are observed for the different strengths of mooring lines.

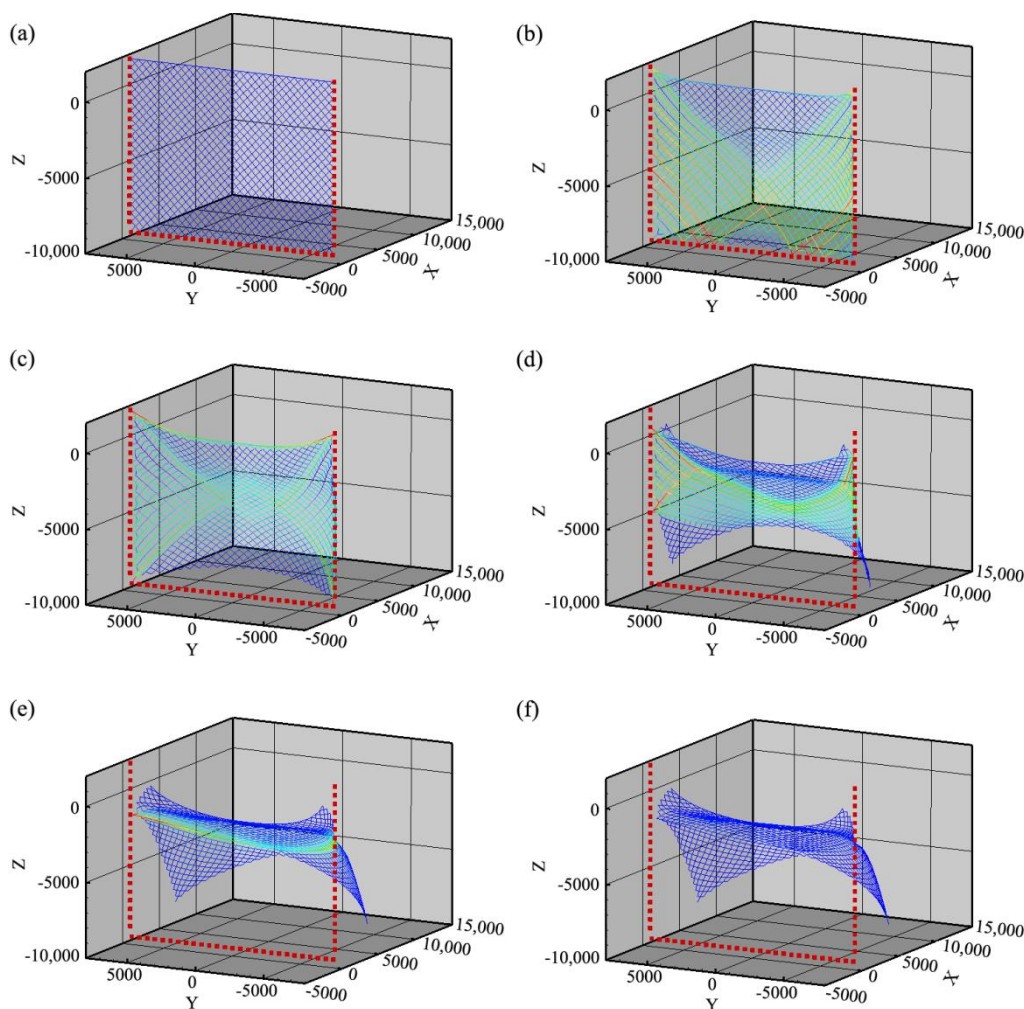

**Figure 15.** (**a–f**) Damage process of the mooring system for the trash-blocking nets when $T_m$ = 3.0 kN.

When the strength of mooring lines ($T_m$) is 3.0 kN, the bottom mooring lines break first, and then the side mooring lines start to break. When the strength of mooring lines ($T_m$) is 4.0 kN, the top part of the side mooring lines breaks first, and then all of the side mooring lines are broken; finally, the fracture of the bottom mooring lines occurs. In general, the whole mooring system can be destroyed when one mooring line starts to break, and the damage mode of mooring lines is determined by the strength of mooring lines.

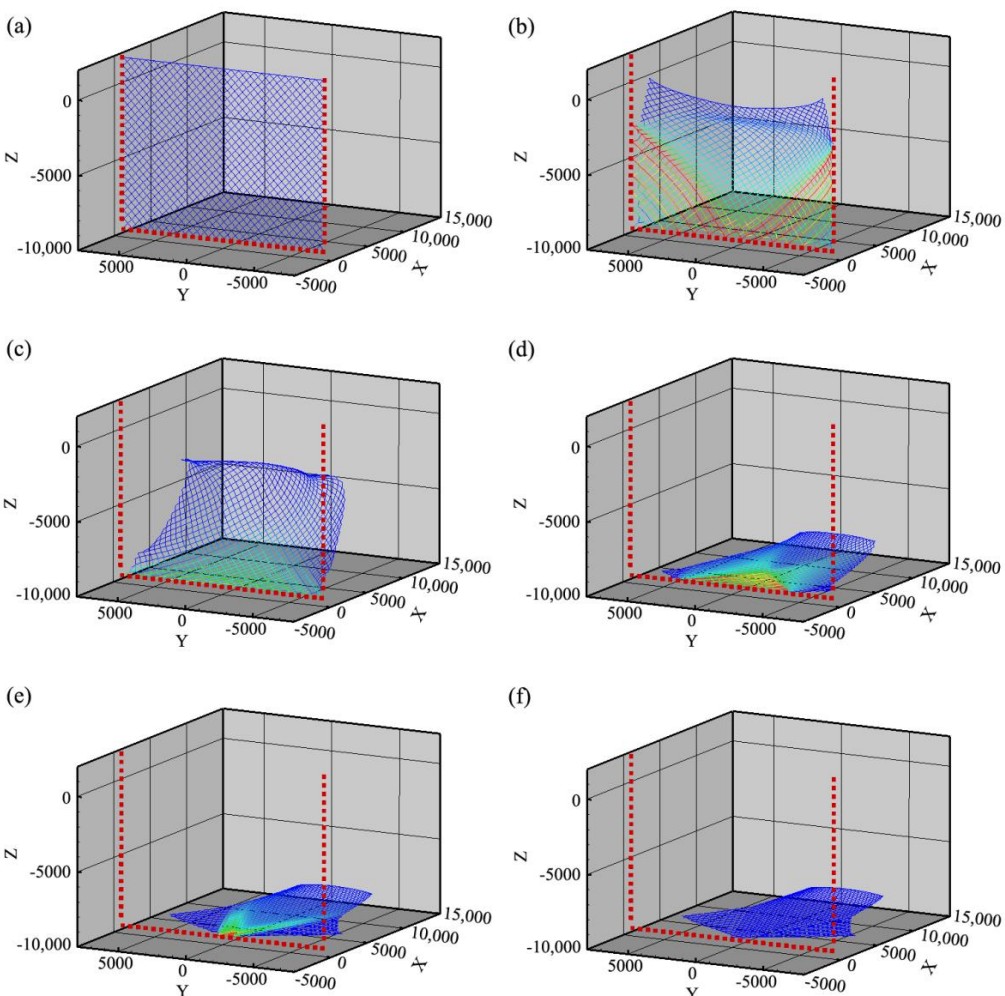

**Figure 16.** (**a**–**f**) Damage process of mooring system for the trash-blocking nets when $T_m = 4.0$ kN.

## 5. Conclusions

A numerical model of the trash-blocking net and mooring system in current is developed based on the lumped mass method. A series of experimental tests are conducted to validate our numerical model. The results indicate that our numerical model is appropriate to calculate the tension force on the trash-blocking net and mooring system.

(1)   The maximum tension force on the trash-blocking nets linearly increases with the increasing width of trash-blocking nets.

(2)   The maximum tension force on the trash-blocking nets significantly decreases with the increase in the horizontal hanging ratio, and it decreases by 87% when the horizontal hanging ratio increases from 0.3535 to 0.9191.

(3)   When the water depth increases from 4.0 m to 10.0 m, the total forces on the side moorings and the bottom moorings decrease by 42.5% and 55.9%, respectively.

(4)   The whole mooring system of the trash-blocking net can be destroyed when one mooring line starts to break. Two damage modes for mooring lines can be observed, and the damage mode of the mooring lines is determined by the strength of mooring lines.

The present study mainly focused on the in-current tension force on nets and mooring systems, and it is suggested that the dynamic response of nets and mooring system in waves be analyzed in the future work.

**Author Contributions:** Conceptualization, Z.J. and T.X.; methodology, T.X.; software, T.W.; validation, B.W. and C.M.; writing—original draft preparation, Z.J. and K.S.; writing—review and editing, T.W.; visualization, T.X.; supervision, T.X.; funding acquisition, T.X. All authors have read and agreed to the published version of the manuscript.

**Funding:** This research was funded by the National Natural Science Foundation (NSFC), grant numbers 51979037 and 52101334; and the Fundamental Research Funds for the Central Universities, grant number DUT21LK30.

**Institutional Review Board Statement:** Not applicable.

**Informed Consent Statement:** Not applicable.

**Data Availability Statement:** Not applicable.

**Conflicts of Interest:** The authors declare no conflict of interest.

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
