# Peer review of "Numerical Investigation of the Hydrodynamic Behavior of Trash-Blocking Nets for Water Intake Engineering of Nuclear Power Plant"

_fluids, doi:10.3390/fluids7070234_

Round 1

Reviewer 1 Report

The authors have developed a numerical model of the trash blocking net and mooring system in current based on the lumped mass method. A series of experimental tests were conducted by the authors to validate the numerical model. Overall, the topic is interesting and the paper is well written. Few minor modifications need to be incorporated.

1. Conclusions part can be written in tabulated form to easy reference for the readers.

2. The following paper can be cited in the introduction of the manuscript.

Koley, Santanu, and Trilochan Sahoo. "Scattering of oblique waves by permeable vertical flexible membrane wave barriers." Applied Ocean Research 62 (2017): 156-168.

Author Response

-Reviewer 1

The authors have developed a numerical model of the trash blocking net and mooring system in current based on the lumped mass method. A series of experimental tests were conducted by the authors to validate the numerical model. Overall, the topic is interesting and the paper is well written. Few minor modifications need to be incorporated.

Reply: Thank you for your comments. The manuscript has been revised seriously according to the reviewers’ comments.

  1. Conclusions part can be written in tabulated form to easy reference for the readers.

Reply: Conclusions part has been rewritten in tabulated form in the revised manuscript.

  1. The following paper can be cited in the introduction of the manuscript.

Koley, Santanu, and Trilochan Sahoo. "Scattering of oblique waves by permeable vertical flexible membrane wave barriers." Applied Ocean Research 62 (2017): 156-168.

Reply: The above paper has been cited in the introduction of the revised manuscript.

Reviewer 2 Report

The work presented by Jiang et al. is of interest, it is well written and the calculations presented contribute to improve the design of the trash blocking nets studied, as well as providing a methodology that allows the study of various considerations in the design of these nets.

The experimental validation of the numerical modelling is where the greatest weaknesses is observed. This verification is limited only to the values in the mooring system for a very specific case and with loads very far from the real ones. No validation is presented for other parts of the network, and this, according to this reviewer, reduces the validity of the work. It is recommended to improve the experimental validation section of the presented calculation methodology.

For instance, the Figure 5 shows the deformation and tension distribution of trash blocking net in current but these values presented are from numerical simulation, i.e. calculated, and are not compared with others experimental and measured, as only the load cells attached at the mooring lines register experimental measurements. So, Figure 5 has only been experimentally confirmed in the two mooring points. This is an important shortcut of the present work. The results are not sufficiently confirmed by experimentally obtained values.

For this, it is recommended that the Section 3. Experimental validation should be extensively improved and extended in order to justify the approximation of the numerical model to the experimentally observed values.

Author Response

-Reviewer 2

The work presented by Jiang et al. is of interest, it is well written and the calculations presented contribute to improve the design of the trash blocking nets studied, as well as providing a methodology that allows the study of various considerations in the design of these nets.

Reply: Thank you for your comments. The manuscript has been revised seriously according to the reviewers’ comments.

The experimental validation of the numerical modelling is where the greatest weaknesses is observed. This verification is limited only to the values in the mooring system for a very specific case and with loads very far from the real ones. No validation is presented for other parts of the network, and this, according to this reviewer, reduces the validity of the work. It is recommended to improve the experimental validation section of the presented calculation methodology.

For instance, the Figure 5 shows the deformation and tension distribution of trash blocking net in current but these values presented are from numerical simulation, i.e. calculated, and are not compared with others experimental and measured, as only the load cells attached at the mooring lines register experimental measurements. So, Figure 5 has only been experimentally confirmed in the two mooring points. This is an important shortcut of the present work. The results are not sufficiently confirmed by experimentally obtained values.

For this, it is recommended that the Section 3. Experimental validation should be extensively improved and extended in order to justify the approximation of the numerical model to the experimentally observed values.

Reply: It is difficult to measure the tension force on the nets accurately, and the tension force on mooring lines is determined by the distribution of tension force on the nets. Therefore, the tension force on mooring lines for a net panel with eight-point mooring system is measured to further validate our numerical model of nets in current in the revised manuscript. In addition, the distribution characteristic of tension force on nets with eight-point mooring system is also compared with the results in Suzuki et al.

Reviewer 3 Report

REVIEW

on article

Numerical investigation of the hydrodynamic behavior of trash

 blocking nets for water intake engineering of nuclear power

 plant

Zhen-Qiang Jiang, Tong-Yan Wang, Bin Wang , Tiao-Jian Xu,

Chang-Lei Ma and Kan-Min Shen

SUMMARY

The article submitted for review is devoted to the current topic: "Numerical investigation of the hydrodynamic behavior of trash blocking nets for water intake engineering of a nuclear power plant." The topic that the authors touched upon is a rather urgent problem for the modern nuclear industry, and environmental friendliness and the functioning of these facilities are at the forefront. The article uses interesting methodological instruments, presents numerical models, analyzes the current situation in the industry, compares analytical data with experimental data, and obtains several important results.

The main result of the study is the fact that the maximum tensile forces of the debris blocking net increase linearly with the width of the debris block nets and decrease significantly with the increase in the horizontal suspension factor of the debris block nets.

Thus, the article has a certain level of practical significance and novelty.

The references list comprises 21 sources.

The study is original and could be published in the Journal Fluids. However, several important points need to be corrected first. The reviewer's comments are presented below.

COMMENTS

1.    The Abstract describes the current state of the issue in sufficient detail, but does not formulate the research problem. A statement of the problem to be solved and what the study is aimed at should be added.

2.    In general, the Abstract is presented quite briefly; more attention should be paid to the main scientific points of the article. The practical, applied significance of the study is presented in detail, but I would like to see a large share of the scientific novelty presented in the Abstract. Information about this should be added.

3.    There are only four phrases in the keywords. To better identify the study, to understand its place in the modern scientific system, the number of keywords and phrases should be increased to 6-7. Then the study will be more searchable in scientific databases.

4.    The Introduction was done by the authors at a good level. However, a few sources studied is reviewed. The authors should analyze more sources on the problems of waste collection at nuclear power plants. There is a large amount of material on this topic in the scientific literature, so the number of sources in the "Introduction" should be increased from 15 to 20-25.

5.    The section "Introduction" should be supplemented with clear formulations of practical significance and scientific novelty.

6.    Section 2 begins with the formulation and definition of the debris blocking grid and how it works. At the same time, on lines 106-108, the mass accumulation method is indicated, but a slightly more extended preamble should be given with an explanation of why such a method was chosen and what caused its choice.

7.    The beginning of subsection 2.1 is Figure 2, but I would also like to see some kind of preamble so that the article looks more organic and the transitions are smoother.

8.    The description of the method of concentrated mass, which is presented at the beginning of subsection 2.1, should be moved a little further, first giving some preamble to the paragraph.

9.    There are very small characters in Figure 2, consideration should be given to increasing the size of the figure or presenting it in a higher resolution.

10.  Formulas 3 and 4 need a little more detailed description due to the fact that only a reference to a previous study is provided for the application of these formulas. A text transition should be added that justifies the relationship between the applied formulas and the study based on which they are given with the study conducted by the authors of the article.

11.  Subsection 2.2 starts with Figure 3, which is methodologically not entirely correct, some preamble should be added to the figure.

12.  Figure 4 contains poorly distinguishable characters, too small, which should also be explained. In addition, the photograph shown in Figure 4a is of rather poor quality and may need to be replaced with a higher-quality, high-resolution photograph. At the same time, it should be clarified: is this photo taken by the authors or taken from some source?

13.  The lack of interpretation of Figure 5 after the figure itself is striking. In addition, in the text the reference to this figure is located quite far from the figure itself, which is incorrect, and this part of section 3 should be corrected.

14.  Figure 6 is made in a bright color scheme. Perhaps a different color scheme should be used for a better perception by readers.

15.  Figures 7 and 8 are presented as bar graphs, however, these figures should be supplemented with more interpretation to reflect the dependencies that these figures should illustrate. In addition, Figure 7 after position 17 on the abscissa is absolutely unreadable and it will not be clear to the reader what is shown in the indicated segment of the abscissa.

16.  Figure 10 also shows the distribution of tension on the nets that block debris at various suspension coefficients, and the color scheme was chosen, in the opinion of the reviewer, not entirely well. Perhaps other colors should be added for better perception by readers.

17.  Tables 3 and 4 in subsections 4.3 and 4.4, respectively, look uninformative and need some interpretation or possible reformatting. However, this is at the discretion of the authors.

18.  Figures 11 and 12, running in a row, are a low quality, which occupy a large part of the sheet and are not connected by any unifying textual interpretations. At the same time, there is no conclusion after them and a smooth transition from the “Discussion” section to the “Conclusion” section should be supplemented. I recommend authors rebuild the Figures for more readable aspect.

19.  In addition, "Discussion" is one of the most important sections and needs a more detailed comparison of the results of the authors with the results obtained earlier by other authors.

20.  The "Conclusions" section is too concise and needs some work in terms of a more detailed presentation of the results. In addition, the prospects, and vectors for the development of the study should be added.

21.  The list of references contains only 21 sources, some of which were completed quite a long time ago, more than 5 years ago. To better reflect the scientific novelty, the list of used sources should be supplemented to at least 25-30 and preference should be given to sources published over the past 5 years.

22. Please, check all figures for quality. The journal requires 1000 pix for the shorten side and 300 DPI resolution.

23.  The general conclusion of the reviewer is that the study is promising, but it is necessary to correct the identified comments and submit the article for re-review. In addition, some editorial changes and English language checking are needed.

Author Response

-Reviewer 3

SUMMARY

The article submitted for review is devoted to the current topic: "Numerical investigation of the hydrodynamic behavior of trash blocking nets for water intake engineering of a nuclear power plant." The topic that the authors touched upon is a rather urgent problem for the modern nuclear industry, and environmental friendliness and the functioning of these facilities are at the forefront. The article uses interesting methodological instruments, presents numerical models, analyzes the current situation in the industry, compares analytical data with experimental data, and obtains several important results.

The main result of the study is the fact that the maximum tensile forces of the debris blocking net increase linearly with the width of the debris block nets and decrease significantly with the increase in the horizontal suspension factor of the debris block nets.

Thus, the article has a certain level of practical significance and novelty.

The references list comprises 21 sources.

The study is original and could be published in the Journal Fluids. However, several important points need to be corrected first. The reviewer's comments are presented below.

Reply: Thank you for your comments. The manuscript has been revised seriously according to the reviewers’ comments.

COMMENTS

  1. The Abstract describes the current state of the issue in sufficient detail, but does not formulate the research problem. A statement of the problem to be solved and what the study is aimed at should be added.

Reply: A statement of the problem to be solved and the aim of the study have been added in the abstract of the revised manuscript.

  1. In general, the Abstract is presented quite briefly; more attention should be paid to the main scientific points of the article. The practical, applied significance of the study is presented in detail, but I would like to see a large share of the scientific novelty presented in the Abstract. Information about this should be added.

Reply: The scientific novelty has been added in the abstract of the revised manuscript.

  1. There are only four phrases in the keywords. To better identify the study, to understand its place in the modern scientific system, the number of keywords and phrases should be increased to 6-7. Then the study will be more searchable in scientific databases.

Reply: The number of keywords and phrases has been increased to 6 in the revised manuscript.

  1. The Introduction was done by the authors at a good level. However, a few sources studied is reviewed. The authors should analyze more sources on the problems of waste collection at nuclear power plants. There is a large amount of material on this topic in the scientific literature, so the number of sources in the "Introduction" should be increased from 15 to 20-25.

Reply: More source of waste collection at nuclear power plants has been added in the introduction of the revised manuscript.

  1. The section "Introduction" should be supplemented with clear formulations of practical significance and scientific novelty.

Reply: The formulations of practical significance and scientific novelty have been added in the introduction of the revised manuscript.

  1. Section 2 begins with the formulation and definition of the debris blocking grid and how it works. At the same time, on lines 106-108, the mass accumulation method is indicated, but a slightly more extended preamble should be given with an explanation of why such a method was chosen and what caused its choice.
  2. The beginning of subsection 2.1 is Figure 2, but I would also like to see some kind of preamble so that the article looks more organic and the transitions are smoother.
  3. The description of the method of concentrated mass, which is presented at the beginning of subsection 2.1, should be moved a little further, first giving some preamble to the paragraph.

Reply: Some explanation of why such a method was chosen has been added in the revised manuscript.

  1. There are very small characters in Figure 2, consideration should be given to increasing the size of the figure or presenting it in a higher resolution.

Reply: The size of characters in Figure 2 has been increased in the revised manuscript.

  1. Formulas 3 and 4 need a little more detailed description due to the fact that only a reference to a previous study is provided for the application of these formulas. A text transition should be added that justifies the relationship between the applied formulas and the study based on which they are given with the study conducted by the authors of the article.

Reply: More reference for the application of Formulas 3 and 4 and a text transition have been added in the revised manuscript.

  1. Subsection 2.2 starts with Figure 3, which is methodologically not entirely correct, some preamble should be added to the figure.

Reply: Some description of the permeable characteristic of nets has been given in the revised manuscript.

  1. Figure 4 contains poorly distinguishable characters, too small, which should also be explained. In addition, the photograph shown in Figure 4a is of rather poor quality and may need to be replaced with a higher-quality, high-resolution photograph. At the same time, it should be clarified: is this photo taken by the authors or taken from some source?

Reply: The size of characters has been increased and a higher-quality, high resolution photograph taken by authors has been given in Figure4 in the revised manuscript.

  1. The lack of interpretation of Figure 5 after the figure itself is striking. In addition, in the text the reference to this figure is located quite far from the figure itself, which is incorrect, and this part of section 3 should be corrected.

Reply: The reference to Figure 5 has been moved close to the figure itself in the revised manuscript.

  1. Figure 6 is made in a bright color scheme. Perhaps a different color scheme should be used for a better perception by readers.

Reply: The authors have attempted to use a different color scheme; however, we cannot obtain a better color scheme. In general, the red color is used to represent the high tension force, and the blue color is used to represent the low tension force.

  1. Figures 7 and 8 are presented as bar graphs, however, these figures should be supplemented with more interpretation to reflect the dependencies that these figures should illustrate. In addition, Figure 7 after position 17 on the abscissa is absolutely unreadable and it will not be clear to the reader what is shown in the indicated segment of the abscissa.

Reply: More interpretation has been added for Figures 11 and 12 (the original Figures 7 and 8) in the revised manuscript. In addition, Figures 11 and 12 (the original Figures 7 and 8) have been modified to ensure it to be clear to the reader in the revised manuscript.

  1. Figure 10 also shows the distribution of tension on the nets that block debris at various suspension coefficients, and the color scheme was chosen, in the opinion of the reviewer, not entirely well. Perhaps other colors should be added for better perception by readers.

Reply: The authors have attempted to use a different color scheme; however, we cannot obtain a better color scheme. In general, the red color is used to represent the high tension force, and the blue color is used to represent the low tension force.

  1. Tables 3 and 4 in subsections 4.3 and 4.4, respectively, look uninformative and need some interpretation or possible reformatting. However, this is at the discretion of the authors.

Reply: Some interpretation and reformatting have been added for Table 3 and 4 in the revised manuscript.

  1. Figures 11 and 12, running in a row, are a low quality, which occupy a large part of the sheet and are not connected by any unifying textual interpretations. At the same time, there is no conclusion after them and a smooth transition from the “Discussion” section to the “Conclusion” section should be supplemented. I recommend authors rebuild the Figures for more readable aspect.

Reply: Figures 15 and 16 (the original Figures 11 and 12) have been rebuilt and some conclusion after them has been supplemented in the revised manuscript

  1. In addition, "Discussion" is one of the most important sections and needs a more detailed comparison of the results of the authors with the results obtained earlier by other authors.

Reply: The comparison of our results with the results obtained earlier by other authors has been given in Section 3.2 in the revised manuscript.

  1. The "Conclusions" section is too concise and needs some work in terms of a more detailed presentation of the results. In addition, the prospects, and vectors for the development of the study should be added.

Reply: More detailed presentation of the results, and the prospects and vectors for the development of the study have been added in the “Conclusion” section of the revised manuscript.

  1. The list of references contains only 21 sources, some of which were completed quite a long time ago, more than 5 years ago. To better reflect the scientific novelty, the list of used sources should be supplemented to at least 25-30 and preference should be given to sources published over the past 5 years.

Reply: Some references have been added in the revised manuscript.

  1. Please, check all figures for quality. The journal requires 1000 pix for the shorten side and 300 DPI resolution.

Reply: All figures have been checked in the revised manuscript.

  1. The general conclusion of the reviewer is that the study is promising, but it is necessary to correct the identified comments and submit the article for re-review. In addition, some editorial changes and English language checking are needed.

Reply: The manuscript has been modified according to the reviewers’ comments, and the editorial changes and English language checking have also been performed in the revised manuscript.

Round 2

Reviewer 1 Report

Since the authors have modified the paper in an appropriate manner, the paper can be accepted in the present form

Reviewer 2 Report

Authors considered all the comments and adressed these in a proper way.

Reviewer 3 Report

All my comments were considered and appropriate corrections were done in the article's text. I recommend the article for publishing.